# Association between pharmaceutical modulation of oestrogen in postmenopausal women in Sweden and death due to COVID-19: a cohort study

Malin Sund,[1,2] Osvaldo Fonseca-Rodríguez,[3] Andreas Josefsson,[1,4,5] Karin Welen,[4] Anne-Marie Fors Connolly  [3]

¹Department of Surgical and Perioperative Sciences, Umeå University Faculty of Medicine, Umeå, Sweden
²Department of Surgery, Univerisity of Helsinki and Helsinki University Hospital, Helsinki, Finland
³Department of Clinical Microbiology, Umeå University Faculty of Medicine, Umeå, Sweden
⁴Department of Urology, University of Gothenburg, Gothenburg, Sweden
⁵Wallenberg Center for Molecular Medicine, Umeå University, Umeå, Sweden

**Correspondence to**
Dr Anne-Marie Fors Connolly;
anne-marie.fors.connolly@umu.se

## ABSTRACT

**Objective** Determine whether augmentation of oestrogen in postmenopausal women decreases the risk of death following COVID-19.

**Design** Nationwide registry-based study in Sweden based on registries from the Swedish Public Health Agency (all individuals who tested positive for SARS-CoV-2); Statistics Sweden (socioeconomical variables) and the National Board of Health and Welfare (causes of death).

**Participants** Postmenopausal women between 50 and 80 years of age with verified COVID-19.

**Interventions** Pharmaceutical modulation of oestrogen as defined by (1) women with previously diagnosed breast cancer and receiving endocrine therapy (decreased systemic oestrogen levels); (2) women receiving hormone replacement therapy (increased systemic oestrogen levels) and (3) a control group not fulfilling requirements for group 1 or 2 (postmenopausal oestrogen levels). Adjustments were made for potential confounders such as age, annual disposable income (richest group as the reference category), highest level of education (primary, secondary and tertiary (reference)) and the weighted Charlson Comorbidity Index (wCCI).

**Primary outcome measure** Death following COVID-19.

**Results** From a nationwide cohort consisting of 49 853 women diagnosed with COVID-19 between 4 February and 14 September 2020 in Sweden, 16 693 were between 50 and 80 years of age. We included 14 685 women in the study with 11 923 (81%) in the control group, 227 (2%) women in group 1 and 2535 (17%) women in group 2. The unadjusted ORs for death following COVID-19 were 2.35 (95% CI 1.51 to 3.65) for group 1 and 0.45 (0.34 to 0.6) for group 2. Only the adjusted OR for death remained significant for group 2 with OR 0.47 (0.34 to 0.63). Absolute risk of death was 4.6% for the control group vs 10.1% and 2.1%, for the decreased and increased oestrogen groups, respectively. The risk of death due to COVID-19 was significantly associated with: age, OR 1.15 (1.14 to 1.17); annual income, poorest 2.79 (1.96 to 3.97), poor 2.43 (91.71 to 3.46) and middle 1.64 (1.11 to 2.41); and education (primary 1.4 (1.07 to 1.81)) and wCCI 1.13 (1.1 to 1.16).

**Conclusions** Oestrogen supplementation in postmenopausal women is associated with a decreased risk of dying from COVID-19 in this nationwide cohort study. These findings are limited by the retrospective and

non-randomised design. Further randomised intervention trials are warranted.

## Strengths and limitations of this study

► This study is based on all diagnosed patients with COVID-19 in Sweden between 1 February and 14 September 2020.

► Swedish registry data are well validated and due to historical registry data and cross-linkage with the registries of Statistics Sweden, the confounding and/or effect-modifying effects of socioeconomic variables and comorbidities could be adjusted for.

► Information regarding compliance to pharmaceutical modulation of oestrogen is missing.

► Information about the exact duration of the postmenopausal hormone therapy was not available in the dataset.

► Circulating oestrogen levels are not measured.

## INTRODUCTION

The COVID-19 pandemic has swept across the globe causing enormous strain on societies and healthcare systems. Although women are infected, they appear to be protected from poor outcomes when compared with men even after adjustment for confounding risk factors.[1,2] Similar epidemiological findings have also been described for SARS-CoV and Middle East Respiratory Syndrome coronavirus (MERS-CoV) infections.[3–5] This implies biological differences between the sexes in terms of sensitivity to severe COVID-19, and oestrogen has been identified as a potential therapeutic candidate.

The majority of patients with breast cancer (BC) have oestrogen receptor-positive cancer[6] and are usually given adjuvant endocrine therapy after surgery in order to reduce the risk of cancer recurrences, leading to *reduced* systemic oestrogen levels. On the other hand, systemic oestrogen levels are *augmented* in

women taking postmenopausal hormone therapy (HT) to relieve menopausal symptoms.[7] In a nationwide cohort, we used the opposing effects of endocrine therapy in patients with BC and women taking postmenopausal HT in modulating systemic oestrogen levels as a model to test the hypothesis that increased oestrogen levels are protective towards COVID-19 death.

## MATERIALS AND METHODS
### Patient and public involvement
All data from the Swedish registries were pseudonymised and therefore patients were not involved in the study.

### Participants and sources of data
The personal identification numbers from all individuals with diagnosed COVID-19 in Sweden (SmiNet) between 1 February and 14 September 2020 were cross-linked with the LISA Register (Longitudinal integrated database for health insurance and labour market studies) administered by Statistics Sweden; and the following healthcare registers administered by the Swedish National Board of Health and Welfare: patient, cancer, prescribed pharmaceutical and causes of death. Postmenopausal women 50–80 years of age were stratified into three groups as follows: group 1, the *decreased oestrogen group,* included patients with BC as identified by the International Classification of Diseases (ICD) version 10 code C50, and the following treatments: tamoxifen or fulvestrant (anatomical therapeutic chemical (ATC): L02BA01, L02BA03) or an aromatase inhibitor (ATC L02BG03, L02BG04 and L02BG06). Group 2, the *augmented oestrogen group,* included those patients treated with drugs classified

as postmenopausal HT (ATC codes G03CA03, G03CA04, G03CC07, G03CX01, G03FA and G03FB). All ATC codes for groups 1 and 2 were identified from the Prescribed Pharmaceutical Register with at least two consecutive withdrawals and at least one during the period extending from 1 July 2019 to the latest date. Group 3, the *native oestrogen (control) group,* included patients with no BC diagnosis and no prescription of the above-mentioned pharmaceuticals at any time during 2019 and 2020.

### Outcome, confounders and effect modifiers
The outcome was death due to COVID-19, as identified by the ICD-10 code U07, as the main or contributing cause of death from the Cause of Death Register. Potential confounders and effect modifiers were included in the model and consisted of the weighted Charlson Comorbidity Index (wCCI),[8] age at COVID-19 diagnosis, income (divided into quintiles with the richest group as the reference) and education (primary, secondary and tertiary, which served as the reference). The wCCI was calculated using the Patient and Cancer Registers, up to 2 months prior to the COVID-19 date in order not to include complications due to COVID-19 as a comorbidity. If there was no information regarding diagnosis codes required for wCCI scoring, the individual was assigned a wCCI of zero. Information regarding income and education was retrieved from the LISA Register.

### Statistical methods
The distributions of continuous and categorical variables in the three groups were tested using Analysis of Variance (ANOVA) and the $\chi^2$ test, respectively. Each variable was then analysed with univariate logistic regression models, followed

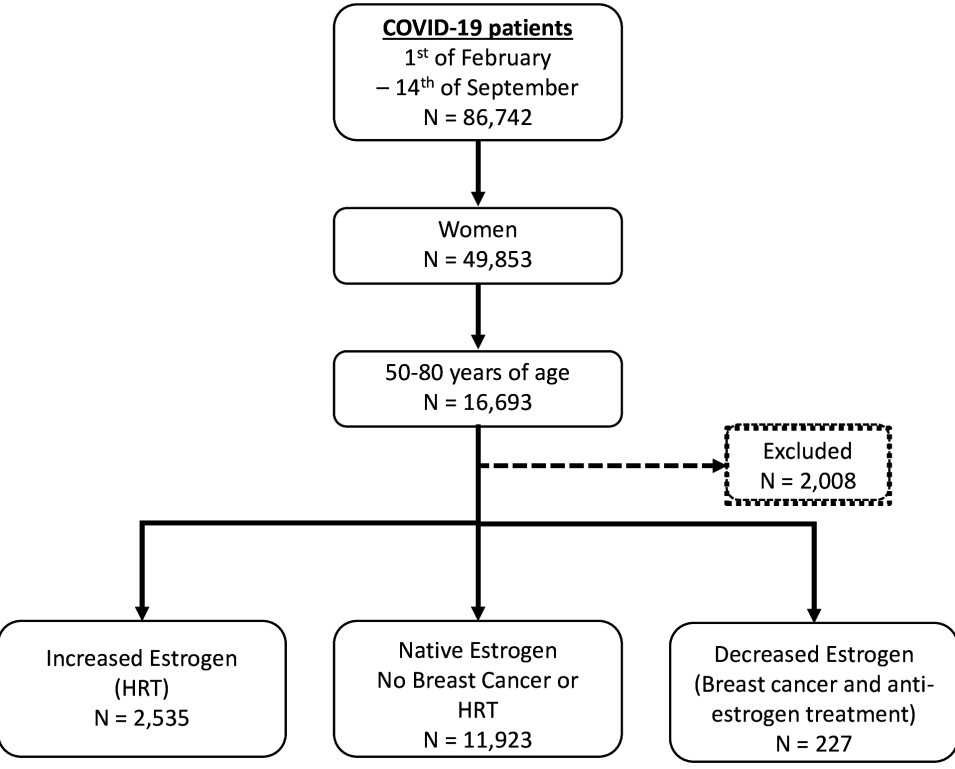

**Figure 1**  Flow chart of the study. HRT, hormone replacement therapy.

**Table 1** Characteristics of the study population

| Variable | | Native oestrogen (control group) | Decreased oestrogen (group 1) | Augmented oestrogen (group 2) | P value |
|---|---|---|---|---|---|
| Total N (%) | | 11 923 (81.2) | 227 (1.5) | 2535 (17.3) | |
| Deaths | No | 11 377 (95.4) | 204 (89.9) | 2481 (97.9) | <0.001 |
| | Yes | 546 (4.6) | 23 (10.1) | 54 (2.1) | |
| Age | Mean (SD) | 61.2 (8.3) | 64.4 (8.9) | 60.9 (7.7) | <0.001 |
| wCCI | Mean (SD) | 1.4 (2.4) | 5.0 (3.3) | 1.6 (2.5) | <0.001 |
| Income quintiles, n (%) | Richest | 3422 (28.7) | 64 (28.2) | 937 (37.0) | <0.001 |
| | Rich | 2743 (23.0) | 42 (18.5) | 605 (23.9) | |
| | Middle | 2120 (17.8) | 35 (15.4) | 404 (15.9) | |
| | Poor | 1703 (14.3) | 47 (20.7) | 334 (13.2) | |
| | Poorest | 1903 (16.0) | 39 (17.2) | 253 (10.0) | |
| | Missing | 32 (0.3) | 0 (0) | 2 (0.1) | |
| Education, n (%) | Tertiary | 4186 (35.1) | 82 (36.1) | 1074 (42.4) | <0.001 |
| | Secondary | 5609 (47.0) | 97 (42.7) | 1150 (45.4) | |
| | Primary | 1882 (15.8) | 45 (19.8) | 290 (11.4) | |
| | Missing | 246 (2.1) | 3 (1.3) | 21 (0.8) | |

wCCI, weighted Charlson Comorbidity Index.

by a multivariable regression model to compare the control group with groups 1 and 2, respectively, and adjusting for confounders. The present study evaluates specific outcomes, and the OR and p values are adjusted for relevant confounders using the multivariate logistic regression model and there was no need to further adjust using the Bonferroni/Benjamini/false discovery rate (FDR) approach. Descriptive analyses and logistic regression models were performed using R statistical software V.4.0.2, using the *finalfit* package 1.0.2.

## RESULTS
### Participants
During the study period, a total of 49 853 women of all ages were diagnosed with COVID-19 in Sweden, and a total of 14 685 women aged 50–80 years were included in our study (figure 1). Characteristics of these groups are shown in table 1. Individuals with decreased oestrogen

due to adjuvant endocrine therapy for BC (group 1) were older with a higher comorbidity index. A larger proportion of women in group 2 (increased levels of systemic oestrogen due to postmenopausal HT) had high income and a tertiary level of education (table 1).

### Oestrogen augmentation protects against death due to COVID-19
Pharmaceutically decreasing systemic oestrogen levels increased the odds of dying due to COVID-19 (group 1: OR 2.35, 95% CI 1.51 to 3.65), but following adjustment for confounders this association was no longer significant (figure 2). Interestingly, augmentation of systemic oestrogen levels decreased the odds of dying due to COVID-19, with OR 0.45 (95% CI 0.34 to 0.6), and this result remained significant even after adjustment for confounders (OR 0.47, 95% CI 0.34 to 0.63). The absolute risk of dying was 4.6% for the control group vs 10.1%

| | Dead | Alive | Crude OR(95%CI) | | Adjusted OR(95%CI) | |
|---|---|---|---|---|---|---|
| **Groups, n(%)** | | | | | | |
| Native oestrogen (control) | 546 (4.6) | 11377 (95.4) | | 1(ref) | | 1(ref) |
| Oestrogen decrease (group 1) | 23 (10.1) | 204 (89.9) | | 2.35 (1.51-3.65)*** | | 1.21 (0.74-1.98) |
| Augmented oestrogen (group 2) | 54 (2.1) | 2481 (97.9) | | 0.45 (0.34-0.60)*** | | 0.47 (0.34-0.63)*** |
| **Age in years** | | | | | | |
| Mean(SD) | 73.2 (6.4) | 60.7 (7.9) | | 1.19 (1.18-1.21)*** | | 1.15 (1.14-1.17)*** |
| **wCCI** | | | | | | |
| Mean(SD) | 3.8 (3.1) | 1.4 (2.4) | | 1.27 (1.24-1.30)*** | | 1.13 (1.10-1.16)*** |
| **Income quintile, n(%)** | | | | | | |
| Richest | 47 (1.1) | 4376 (98.9) | | 1(ref) | | 1(ref) |
| Rich | 44 (1.3) | 3346 (98.7) | | 1.22 (0.81-1.85) | | 1.14 (0.74-1.74) |
| Middle | 75 (2.9) | 2484 (97.1) | | 2.81 (1.95-4.06)*** | | 1.64 (1.11-2.42)* |
| Poor | 198 (9.5) | 1886 (90.5) | | 9.77 (7.08-13.50)*** | | 2.44 (1.71-3.47)*** |
| Poorest | 258 (11.8) | 1937 (88.2) | | 12.40 (9.05-17.00)*** | | 2.79 (1.96-3.98)*** |
| **Education, n(%)** | | | | | | |
| Tertiary | 114 (2.1) | 5228 (97.9) | | 1(ref) | | 1(ref) |
| Secondary | 241 (3.5) | 6615 (96.5) | | 1.67 (1.33-2.09)*** | | 1.15 (0.90-1.47) |
| Primary | 229 (10.3) | 1988 (89.7) | | 5.28 (4.20-6.65)*** | | 1.40 (1.07-1.81)* |

**Figure 2** Oestrogen augmentation is associated with decreased odds of dying due to COVID-19. Crude and adjusted logistic regression models. Statistical significance: *p<0.05, **p<0.01, ***p<0.001. wCCI, weighted Charlson Comorbidity Index.

and 2.1% for the groups with decreased and increased oestrogen, respectively.

As expected, higher age and wCCI increased the odds of dying due to COVID-19. For every year increase in age, the odds of dying were 1.15 (95% CI 1.14 to 1.17), and for every increase in wCCI the odds of dying were 1.13 (95% CI 1.10 to 1.16) (figure 2). Furthermore, low income and having only primary education were also factors that increased the odds of dying due to COVID-19 (figure 2).

## DISCUSSION
### Principal findings
The major finding of this nationwide registry-based study is that pharmaceutical augmentation of oestrogen levels is associated with decreased odds of death due to COVID-19 in postmenopausal women.

### Comparison with related studies
There are several possible biological explanations for the lower risk experienced by women. These include mechanisms directly involved in viral internalisation and reproduction, where oestrogen has been shown to decrease expression of vital proteins such as ACE2 and TMPRSS2,[9–11] inherent sex-linked differences in the immune system, and direct oestrogen effects.[12] As an example, Kalidhindi *et al* have studied the effect of testosterone and oestrogen on ACE2, a key cell entry for SARS-CoV-2 virus, using in vitro experiments on isolated human airway smooth muscle cells of male and female origin.[13] Most interestingly, they show that cells exposed to oestrogen and testosterone behave differently, as testosterone significantly upregulates ACE2 expression in cells from both sexes, whereas oestrogen downregulates ACE2.[13] ACE2 expression and differences in its expression in relation to sex could also be linked to the higher mortality in relation to hypertension, venous thromboembolism and SARS-CoV-2 infection between men and women.[14] Our findings are also supported by in vitro studies where 17β-oestradiol treatment reduced SARS-CoV-2 viral load.[9] Previous experimental studies in mice on SARS-CoV have, moreover, shown that female mice are less susceptible to infection and that this protection was lost on oophorectomy, thus indicating a direct protective role of oestrogen signalling.[15] Furthermore, Barh *et al*, using a multiomics approach on SARS-CoV-2-infected host interactome, proteome, transcriptome and bibliome datasets, demonstrated that oestrogen modulation could be a potential therapeutic option in COVID-19.[16] Our results are in line with those by Seeland *et al* using real-world evidence from multiple institutions and the TriNetX platform. They found by using propensity score matched analysis of data for women aged 50 years and above with COVID-19 (n=439) that there was a survival benefit for oestradiol hormone users versus non-users (OR 0.33 (95% CI 0.18 to 0.62)). Although based on a large real-world dataset, the risk of selection bias was more difficult to discern since the cohort was neither population based

nor adjusted for central confounders although likely mitigated by the propensity score matched analysis.[17] In our study, the effect of increased systemic oestrogen levels on reducing the risk of COVID-19 death remained significant after adjusting for education level and income, both factors known to influence COVID-19 outcome,[18] further supporting the protective role of oestrogen in women. Adjusting for income and education is important as we have previously shown how these affect the risk of dying due to COVID-19 in Sweden.[19]

The hypothetical inverse, a worsening effect of reduced systemic oestrogen levels in women with a previous BC receiving adjuvant endocrine therapy, was initially significant but not after adjusting for confounders. This population differs from the control group in that they all have been diagnosed with BC and it has been shown that patients, both men and women, with any cancer are harder hit by COVID-19.[20] However, in a previous study, patients with BC were shown to be healthier compared with the background population in terms of ischaemic cardiac disease and CCI,[21] and the wCCI adjustments may therefore overcompensate for this cancer-related vulnerability. Although not significant, a trend towards worse outcome remained and thus a larger population of patients with BC on endocrine therapy is likely needed to verify the finding. Thus, this study cannot exclude an increased risk of death from COVID-19 if systemic oestrogen levels are pharmaceutically decreased.

### Strengths and limitations
The strengths of this study are that this is a nationwide cohort in a country with high COVID-19 incidence using well-validated registry data. A weakness is that the level of oestrogen modulation cannot be exactly measured in each individual, and that the number of women with BC on anti-oestrogen medication ended up being too small to show significance although there was a clear trend. Furthermore, we do not have data on the exact duration of postmenopausal HT for the individuals. The postmenopausal HT group, however, proved large enough to show the clear protective effect. A further limitation is that confounding factors such as body mass index, nutrition and smoking habits are not available in the nationwide registry data.

### Implications and conclusion
This study shows an association between oestrogen levels and COVID-19 death. Consequently, drugs increasing oestrogen levels may have a role in therapeutic efforts to alleviate COVID-19 severity in postmenopausal women and could be studied in randomised control trials.

**Acknowledgements** We would like to thank Wolfgang Lohr for data management and Dr Chloé Jacquet for helping us design figure 2.

**Contributors** MS and KW conceptualised the study. MS and A-MFC designed the study. OF-R and A-MFC prepared the study data. OF-R performed the statistical analysis. MS, OF-R, KW, AJ and A-MFC all contributed to interpretation of the results. MS wrote the first draft of the manuscript. MS, OF-R, KW, AJ and A-MFC

contributed to critical revision of the manuscript. MS, OF-R, KW, AJ and A-MFC approved the final manuscript. A-MFC is the guarantor of this study.

**Funding** This study was funded by: A-MFC—Central ALF-funding, Region Västerbotten (RV-836351), Base unit ALF-funding (RV-939769); Strategic Funding during 2020 from the Department of Clinical Microbiology, Umeå University; Stroke Research in Northern Sweden; and Molecular Infection Medicine Sweden (MIMS). AJ—the Knut and Alice Wallenberg Foundation.

**Competing interests** None declared.

**Patient and public involvement** Patients and/or the public were not involved in the design, or conduct, or reporting, or dissemination plans of this research.

**Patient consent for publication** Not required.

**Ethics approval** Ethical approval was obtained from the Swedish Ethical Review Authority (number 2020-02150).

**Provenance and peer review** Not commissioned; externally peer reviewed.

**Data availability statement** No data are available. The study protocol (R script) is available upon request. The study used secondary registry data that are regulated by the Public Access to Information and Secrecy Act (2009:400) and are protected by strict confidentiality. For the purpose of research though, after formal application to access personal data, the responsible authority can grant access to data, though this is contingent on vetting by the Ethical Review Authority of Sweden, according to the Act (2003:460) concerning the Ethical Review of Research Involving Humans. This means that the aggregated registry data cannot be shared.

**ORCID iD**
Anne-Marie Fors Connolly http://orcid.org/0000-0001-9215-4047

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
