## [Reviewer comments · BMJ Open]

ARTICLE DETAILS

TITLE (PROVISIONAL)	Association between pharmaceutical modulation of oestrogen in postmenopausal women in Sweden with death due to COVID-19 – a cohort study
AUTHORS	Sund, Malin; Fonseca-Rodríguez, Osvaldo; Josefsson, Andreas; Welen, Karin; Fors Connolly, Anne-Marie

VERSION 1 – REVIEW

REVIEWER	Seeland, Ute Charite Universitätsmedizin Berlin
REVIEW RETURNED	28-Jun-2021

GENERAL COMMENTS	Thank you for this interesting study to confirm the findings of positive estradiol effect on women with SARS-Cov2 infection. I have a few suggestions for improving the writing style of the paper and correcting the content under discussion. 1. Abstract: Please give the reader a little more information about the purpose of the study and please do not use a sentence beginning with "if". The purpose of the study is to investigate whether adding oestrogen in the postmenopausal period reduces the risk of death in women with COVID-19 and whether suppressing serum levels of oestrogen increases the rate of death.2. Abstract: Please provide some additional information about this registry. Which country? Recruitment?3. Please improve the writing style and English language of the paper.4. Main outcome measures: You do not need this heading when starting the sentence with the same words. Furthermore, you do not need the explanations in brackets.5. The term hormone replacement therapy should not be used any longer. I prefer to use postmenopausal hormone therapy.6. Why do you need this very short paragraph? „Patients and public involvement statement: All data from the Swedish registries were pseudonymized and therefore patients were not involved in the study.“ Integrate this information to the next paragraph.7. Please improve the writing style with the help of an English native speaker. „The personal identification numbers (PINs) from all diagnosed COVID-19 individuals in Sweden between the 1st of February to 14th of September 2020 were cross-linked with the LISA Register (Longitudinal integrated database for health insurance and labor market studies) administered by Statistics Sweden; and the following healthcare registers administered by the Swedish National Board of Health and Welfare: Patient; Cancer; Prescribed pharmaceutical and Causes of Death....8. Please describe the items included within the weighted Charlson comorbidity index. For the reader it is more comfortable to get an
--

	idea about this index without reading the original paper. 9. Discussion part: „These include mechanisms directly involved in viral internalization and reproduction, where oestrogen has been shown to decrease expression of vital proteins such as ACE2 and TMPRSS2 9 ,10“. These references do not explain the mechanisms completely, however these are important findings with limitations. It is an invitro study and the limitations are described by the authors very well: „We should emphasize that the observed E2-induced reduction of ACE2 mRNA might not necessarily translate into a reduction of ACE2 protein at the cell surface.“ The authors should discuss these findings more as a hint that estradiol regulates ACE2 and TMPRSS2 on different cell levels and this might be important to understand the sexual dimorphism known by the SARS-CoV2 infection. 10. Discussion part: The discussion should be improved concerning the discussion of the results of Seeland et al. This study shows for the first time the benefit for the estradiol hormone-user group vs. the non-user group on fatality in a real world data set of 17 countries in women: „After matching the data of propensity scores for women aged 50+ with COVID-19, there was found to be a benefit for the estradiol hormone-user group vs. the non-user group, as regards an outcome of fatality. Baseline characteristics, including demographics, diagnoses, procedures, and medication, were obtained. Propensity score matching was used to balance cohorts. Propensity scores matched cohorts 1:1 using a nearest neighbor greedy matching algorithm with a caliper of 0.25 times the standard deviation. The primary outcome was defined as death. The OR calculated via logistic regression analysis for the combined outcome variable was 0.33 [0.18, 0.62] and the hazard ratio (HR) was 0.29 [0.11, 0.76] for the estradiol non-user vs. hormone-user group. The risk reduction for fatality from 6.6% (non-user) to 2.3% (user) was statistically significant (p < 0.0001).“Please adapt the discussion and notice the description in the paper about the advantag using real-world evidence (RWE) data affords. 11. Statistical analysis can be done as you suggested because of your special focus to education and income. However, propensity score matching to balance the cohorts would be appropriate in your study as well and might give you the same results without forcing education and income. It is not oblige for your study but would improve the understanding of the paper and the results are more comparable to the results of the first paper by Seeland et al. showing this issue of significant risk reduction for fatality in women using postmenopausal hormone therapy.
--	--

REVIEWER	Sathish, Venkatachalem North Dakota State University, Department of Pharmaceutical Sciences
REVIEW RETURNED	29-Sep-2021

GENERAL COMMENTS	For this cohort study, the authors used good sample size and divided their samples to 3 groups. The main outcome from the nationwide cohort study was: “by giving an estrogen supplementation in post-menopausal women, the risk of dying from COVID-19 is reduced”. However, I do have major concerns:  1. The major outcome of the patients was death following COVID-19 and the exposure was pharmaceutical modulation of estrogen levels. What about the estrogen levels in the patients from group 1 and 2 who survived COVID-19? Some data can be included for more robust results and comparison. 2. The authors considered age, income, and education as
---

	confounding effects in their data. How do the authors suggest an effect of income and education on estrogen levels in females? 3. The sample size is highly variable to make comparisons: 11923 in controls vs. 227 in group 1 vs. 2535 in group 2. How would the authors justify making a comparison with the high variations in sample size? 4. Any specific reason for not including the adjusted p-values after Bonferroni/Benjamini/FDR approach in the data? 5. Some the basic science studies and references should be added in the discussion related to this work. This will strengthen the manuscript. See below for example: PMID: 32966970, PMID: 32996784, PMID: 32331343.
--	---

VERSION 1 – AUTHOR RESPONSE

Reviewer: 1

Dr. Ute Seeland, Charite Universitätsmedizin Berlin Comments to the Author:

Thank you for this interesting study to confirm the findings of positive estradiol effect on women with SARS-Cov2 infection.

I have a few suggestions for improving the writing style of the paper and correcting the content under discussion.

1. Abstract: Please give the reader a little more information about the purpose of the study and please do not use a sentence beginning with "if". The purpose of the study is to investigate whether adding oestrogen in the postmenopausal period reduces the risk of death in women with COVID-19 and whether suppressing serum levels of oestrogen increases the rate of death.

Response: We could not find a sentence beginning with "if". Possibly the reviewer is referring to the objectives sentence: "Determine if oestrogen augmentation decreases the risk of death following COVID-19"?

We followed the BMJ Open guidelines for writing the abstract and for objectives it should be a "clear statement of main study aim and major hypothesis/research question" (from BMJ Open's webpage). However, to accommodate the reviewer comment and still adhere to the BMJ Open's guidelines we have changed to:

Line 29-30: "Determine whether augmentation of oestrogen in post-menopausal women decreases the risk of death following COVID-19."

2. Abstract: Please provide some additional information about this registry. Which country? Recruitment?

Response: Thank you for highlighting this, it will help improve our manuscript.

Line 32-34:

"Design: Nationwide study in Sweden based on registries from The Swedish Public Health Agency; Statistics Sweden (socioeconomical variables) and the National Board of Health and Welfare (Causes of death)."

3. Please improve the writing style and English language of the paper.

Response: The manuscript has undergone professional editing to improve the writing style and English language.

4. Main outcome measures: You do not need this heading when starting the sentence with the same words. Furthermore, you do not need the explanations in brackets.

Response: In order to adhere to BMJ Open's guidelines for structured abstract, we have changed the "Main outcome measures" to "primary outcome measure". We have also added a section called "Interventions".

This now reads:

Line 39-45:

"Interventions: Pharmaceutical modulation of oestrogen as defined by (1) women with breast cancer receiving endocrine therapy (decreased systemic oestrogen levels); (2) postmenopausal hormone therapy (HT; increased systemic oestrogen levels) and (3) a control group (postmenopausal oestrogen levels). Adjustments were made for potential confounders such as age, annual disposable income (richest group as the reference category), highest level

of education (primary, secondary and tertiary (reference)) and the weighted Charlson Comorbidity Index (wCCI).”

5. The term hormone replacement therapy should not be used any longer. I prefer to use postmenopausal hormone therapy.

Response: As requested by the reviewer we have in the revised manuscript used the term “postmenopausal hormone therapy (HT)”.

6. Why do you need this very short paragraph? „Patients and public involvement statement: All data from the Swedish registries were pseudonymized and therefore patients were not involved in the study. “ Integrate this information to the next paragraph.

Response: We have kept this section per the recommendation of the editors. This is included as requested by the BMJ Open guidelines.

7. Please improve the writing style with the help of an English native speaker. „The personal identification numbers (PINs) from all diagnosed COVID-19 individuals in Sweden between the 1st of February to 14th of September 2020 were cross-linked with the LISA Register (Longitudinal integrated database for health insurance and labor market studies) administered by Statistics Sweden; and the following healthcare registers administered by the Swedish National Board of Health and Welfare: Patient; Cancer; Prescribed pharmaceutical and Causes of Death....

Response: The manuscript has undergone professional editing to improve the writing style and English language.

8. Please describe the items included within the weighted Charlson comorbidity index. For the reader it is more comfortable to get an idea about this index without reading the original paper.

Response: The Charlson comorbidity index is a validated index to determine the comorbidity burden for an individual and is commonly used in the literature. The diagnosis codes included in this index spans 15 pages of tables with international classification of disease (ICD) diagnosis codes from the 7th, 8th, 9th and 10th version. The focus of our study is not to study the effect of comorbidities, rather the wCCI is used to determine the burden of comorbidities and to ensure that our results are not confounded by individuals having a higher risk of dying due to increased comorbidity burden.

9. Discussion part: „These include mechanisms directly involved in viral internalization and reproduction, where oestrogen has been shown to decrease expression of vital proteins such as ACE2 and TMPRSS2 9,10“. These references do not explain the mechanisms completely, however these are important findings with limitations. It is an *in vitro* study and the limitations are described by the authors very well: „We should emphasize that the observed E2-induced reduction of ACE2 mRNA might not necessarily translate into a reduction of ACE2 protein at the cell surface.“ The authors should discuss these findings more as a hint that estradiol regulates ACE2 and TMPRSS2 on different cell levels and this might be important to understand the sexual dimorphism known by the SARS-CoV2 infection.

Response: Thank you for commenting on this. We have added a comment on the limitation of *in vitro* based data. We have moreover also expanded this section by adding more detailed data on *in vitro* models where sex steroid hormone-based modelling has been used to study ACE2 expression.

Lines 170-186:

“As an example, Kalidhindi et al have studied the effect of testosterone and oestrogen on ACE2 expression, a key cell entry for SARS-CoV-2 virus, using *in vitro* experiments on isolated human airway smooth muscle cells of male and female origin¹³. Most interestingly, they show that cells exposed to oestrogen and testosterone behave differently, as testosterone significantly upregulates ACE2 expression in cells from both sexes, whereas oestrogen downregulates ACE2¹³. ACE2 expression and differences in its expression in relation to sex could also be linked to the higher mortality in relation to hypertension, venous thromboembolism and SARS-CoV-2 infection between men and women ¹⁴. The observed oestrogen induced reduction of ACE2 expression might however not necessarily translate into reduction of ACE2 protein at the cell surface *in vivo* in all cell types. Our findings are also supported by *in vitro* studies where 17 β -oestradiol treatment reduced SARS-CoV-2 viral load ⁹. Previous experimental studies in mice on SARS-CoV have, moreover, shown that

female mice are less susceptible to infection and that this protection was lost upon oophorectomy, thus indicating a direct protective role of oestrogen signalling 15. Furthermore, Barh et al., using a multiomics approach on SARS-CoV-2-infected host interactome, proteome, transcriptome, and bibliome datasets, demonstrated that oestrogen modulation could be a potential therapeutic option in COVID-19 16.”

10. Discussion part: The discussion should be improved concerning the discussion of the results of Seeland et al. This study shows for the first time the benefit for the estradiol hormone-user group vs. the non-user group on fatality in a real world data set of 17 countries in women: „After matching the data of propensity scores for women aged 50+ with COVID-19, there was found to be a benefit for the estradiol hormone-user group vs. the non-user group, as regards an outcome of fatality. Baseline characteristics, including demographics, diagnoses, procedures, and medication, were obtained. Propensity score matching was used to balance cohorts. Propensity scores matched cohorts 1:1 using a nearest neighbor greedy matching algorithm with a caliper of 0.25 times the standard deviation. The primary outcome was defined as death. The OR calculated via logistic regression analysis for the combined outcome variable was 0.33 [0.18, 0.62] and the hazard ratio (HR) was 0.29 [0.11, 0.76] for the estradiol non-user vs. hormone-user group. The risk reduction for fatality from 6.6% (non-user) to 2.3% (user) was statistically significant ($p < 0.0001$).“ Please adapt the discussion and notice the description in the paper about the advantage using real-world evidence (RWE) data affords.

Response: We apologize if we caused offense to the reviewer for our description of their study, that was not our intention. We are not sure if the reviewer suggests that we include the above suggested text in our discussion?

In order to accommodate this comment, we have expanded the discussion to better highlight the findings in this paper that are in line with those of our manuscript. We have in the revised manuscript also commented on the advantage of real-world evidence and propensity score matched analysis in the Seeland et al paper as requested by reviewer 1.

Lines 186-196:

“Our results are in line with those by Seeland et al using real world evidence from multiple institutions and the TriNetX platform. They found by using propensity score matched analysis of data for women aged 50 and above with COVID-19 (n=439), that there was a survival benefit for oestradiol hormone-users versus non-users (OR 0.33 (95%CI 0.18-0.62)). Although based on a large real-world dataset the risk of selection bias was more difficult to discern since the cohort was neither population-based nor adjusted for central confounders although likely mitigated by the propensity score matched analysis 17. In our study the effect of increased systemic oestrogen levels on reducing the risk of COVID-19 death remained significant after adjusting for education level and income, both factors known to influence COVID-19 outcome 18, further supporting the protective role of oestrogen in women.”

11. Statistical analysis can be done as you suggested because of your special focus to education and income. However, propensity score matching to balance the cohorts would be appropriate in your study as well and might give you the same results without forcing education and income. It is not oblige for your study but would improve the understanding of the paper and the results are more comparable to the results of the first paper by Seeland et al. showing this issue of significant risk reduction for fatality in women using postmenopausal hormone therapy.

Response: We are not sure why the reviewer would like us to change our methodology, and do not agree that this would improve our paper. Our study comprises all individuals in Sweden that tested positive for SARS-CoV-2. From this nationwide cohort consisting of all COVID-19 patients, we select post-menopausal women and further divide these women into groups of native oestrogen (control group) vs augmented oestrogen (women receiving HRT) and decreased oestrogen (women with breast cancer that receive anti-oestrogen treatment). There are no other control individuals, all have tested positive for SARS-CoV-2, and we adjust for relevant confounders as described in the manuscript.

Reviewer: 2

Dr. Venkatachalem Sathish, North Dakota State University Comments to the Author:

For this cohort study, the authors used good sample size and divided their samples to 3 groups. The main outcome from the nationwide cohort study was: “by giving an estrogen supplementation in postmenopausal women, the risk of dying from COVID-19 is reduced”. However, I do have major concerns:

1. The major outcome of the patients was death following COVID-19 and the exposure was pharmaceutical modulation of estrogen levels. What about the estrogen levels in the patients from group 1 and 2 who survived COVID-19? Some data can be included for more robust results and comparison.

Response: It would indeed be very interesting to have information regarding the oestrogen levels. However, this data is based on Swedish registries, where information on oestrogen concentration is not available.

2. The authors considered age, income, and education as confounding effects in their data. How do the authors suggest an effect of income and education on estrogen levels in females?

Response: This is a good point. The reason for including income and education, is that these have been shown to affect the risk of dying due to COVID-19. We recently published a study showing this: “Inequitable impact of infection: social gradients in severe COVID-19 outcomes among all confirmed SARS-CoV-2 cases during the first pandemic wave in Sweden”. Per E Gustafsson, Miguel San Sebastian, Osvaldo Fonseca Rodríguez, Anne-Marie Fors Connolly. Journal of Epidemiology and Community Health. DOI: 10.1136/jech-2021-216778 We are interested in determining the effect of oestrogen modulation on risk of dying due to COVID-19, therefore, we must also remove the confounding effects of income and education on dying due to COVID-19.

3. The sample size is highly variable to make comparisons: 11923 in controls vs. 227 in group 1 vs. 2535 in group 2. How would the authors justify making a comparison with the high variations in sample size?

Response: The sample size is indeed highly variable. The reason for this variation is that all individuals that tested positive for SARS-CoV-2 in Sweden are included, where after we have selected only postmenopausal women and stratified these individuals into three groups. It is not possible to change the composition of these groups, since any that match the criteria for the specified group is included. To remove individuals would introduce bias, and diminish the validity of the study.

4. Any specific reason for not including the adjusted p-values after Bonferroni/Benjamini/FDR approach in the data?

Response: This is an epidemiological cohort study, using the logistic regression to determine the odds ratio of death due to COVID-19 in postmenopausal women that receive treatment that either increase or decrease native oestrogen levels compared to native oestrogen levels in postmenopausal women that do not receive any of these treatments. Our study evaluates specific outcomes, and the odds ratio and p-values are adjusted for relevant confounders using the multivariate logistic regression model.

5. Some the basic science studies and references should be added in the discussion related to this work. This will strengthen the manuscript. See below for example: PMID: 32966970, PMID: 32996784, PMID: 32331343.

Response: Thank you for pointing this out. The discussion has now been expanded with the addition of the above references and a more thorough discussion on the finding of sex steroid hormone effect on ACE2 expression in human cells lines. Furthermore, the reference list is updated to include the three references mentioned by the reviewer.

Lines 170-186:

“As an example, Kalidhindi et al have studied the effect of testosterone and oestrogen on ACE2 expression, a key cell entry for SARS-CoV-2 virus, using in vitro experiments on isolated human airway smooth muscle cells of male and female origin¹³. Most interestingly, they show that cells exposed to oestrogen and testosterone behave differently, as testosterone significantly upregulates ACE2 expression in cells from both sexes, whereas oestrogen downregulates ACE2¹³. ACE2 expression and differences in its expression in relation to sex could also be linked to the higher mortality in relation to hypertension, venous thromboembolism and SARS-CoV-2 infection between men and women ¹⁴. The

observed oestrogen induced reduction of ACE2 expression might however not necessarily translate into a reduction of ACE2 protein at the cell surface in vivo in all cell types. Our findings are also supported by in vitro studies where 17 β -oestradiol treatment reduced SARS-CoV-2 viral load 9. Previous experimental studies in mice on SARS-CoV have, moreover, shown that female mice are less susceptible to infection and that this protection was lost upon oophorectomy, thus indicating a direct protective role of oestrogen signalling 15. Furthermore, Barh et al., using a multiomics approach on SARS-CoV-2-infected host interactome, proteome, transcriptome, and bibliome datasets, demonstrated that oestrogen modulation could be a potential therapeutic option in COVID-19 16.”

VERSION 2 – REVIEW

REVIEWER	Seeland, Ute Charite Universitätsmedizin Berlin
REVIEW RETURNED	07-Nov-2021
GENERAL COMMENTS	The manuscript improved very well after revision.